# Workforce outcomes among substance use peer supports and their contextual determinants: A scoping review protocol

**Justin S. Bell**[1], **Tina Griffin**[2], **Sierra Castedo de Martell**[1], **Emma Sophia Kay**[3], **Mary Hawk**[4], **Michelle Hudson**[1], **Bradley Ray**[5], **Dennis P. Watson**[1]*

1 Lighthouse Institute, Chestnut Health Systems, Chicago, Illinois, United States of America, 2 University Library, University of Illinois Chicago, Chicago, Illinois, United States of America, 3 School of Nursing, The University of Alabama at Birmingham, Birmingham, Alabama, United States of America, 4 School of Public Health, University of Pittsburgh, Pittsburgh, Pennsylvania, United States of America, 5 RTI International, Research Triangle Park, North Carolina, United States of America

☯ These authors contributed equally to this work.
* dpwatson@chestnut.org

**Data Availability Statement:** No datasets were generated or analysed during the current study. All

## Abstract

### Introduction

Peer recovery support services are a promising approach for improving harm reduction, treatment, and recovery-related outcomes for people who have substance use disorders. However, unique difficulties associated with the role may place peer recovery support staff [i.e., peers] at high risk for negative workforce outcomes, including burnout, vicarious trauma, and compassion fatigue.

### Objective

This scoping review protocol aims to describe a proposed effort to review the nature and extent of research evidence on peer workforce outcomes and how these outcomes might differ across service settings. Results of the review described in this protocol will help to answer the following research questions: 1) What is known about workforce-related outcomes for peers working in the substance use field?; 2) What is known about how the structure of work impacts these outcomes?; and 3) How do these outcomes differ by service setting type?

### Methods

A scoping review will be conducted with literature searches conducted in PsycINFO®, [EBSCO],Embase® [EBSCO], CINAHL® [EBSCO], Web of Science™ [Clarivate], and Google Scholar databases for relevant articles discussing US-based research and published in English from 1 January 1999 to 1 August 2023. The proposed review will include peer-reviewed and grey-literature published materials describing the experiences of peers participating in recovery support services and harm reduction efforts across a variety of service settings. Two evaluators will independently review the abstracts and full-text articles. We

relevant data from this study will be made available upon study completion.

**Funding:** This study was supported by funding from the National Institute on Drug Abuse (R33DA045850, MPIs: Watson and McGuire), including funds from the Justice Community Opioid Innovation Network Collaborative cooperative agreement (UG1DA050065; MPIs: Dennis and Grella). The funder provided support in the form of salaries for authors JB, DW, and MH, but did not have any additional role in the study design, data collection and analysis, decision to publish, or preparation of the manuscript. The specific roles of these authors are articulated in the 'author contributions' section. All authors work for universities and health or research organizations that are non-profit entities. There was no additional external funding received for this study.

**Competing interests:** SC is a volunteer board member for a recovery community organization that delivers peer recovery support services. This does not alter our adherence to PLOS ONE policies on sharing data and materials. None of the other authors have competing interests to declare.

will perform a narrative synthesis, summarizing and comparing the results across service settings.

## Expected outputs

Publishing this protocol will help accelerate the identification of critical workforce issues, and bolster the transparency and reporting of the final review. The proposed review will assess the state of the literature on peer workforce-related outcomes and how outcomes might vary by service setting context. Results of the proposed review will be disseminated in peer-reviewed publications and conference presentations. Findings will inform the field regarding future directions to support the emerging peer workforce.

## Trial registration

### Systematic review registration

Submitted to Open Science Framework, August 22nd, 2023.

## Introduction

Peer recovery support services (PRSS) for substance use disorder (SUD) have expanded over the past two decades, and the most recent National Drug Control Strategy recommends continuous development of the PRSS workforce (e.g., peers) [1]. PRSS interventions are also a current research priority of the National Institute on Drug Abuse [2], with several systematic reviews providing support for peer effectiveness related to such outcomes as decreased substance use, increased rates of abstinence-based recovery, strengthened treatment retention, improved provider-participant relationships, and increased treatment satisfaction [3–7]. However, studies suggest workforce-related challenges associated with peer roles, including a lack of role clarity and high potential for burnout and vicarious trauma exposure [8,9]. When considering peer workforce outcomes, it is important to remember that many peers are, themselves, living in recovery or successfully managing their substance use through harm reduction strategies. While previous studies have tended to focus on those certified peer workers or peer recovery coaches who are in active recovery, they have neglected those who might be effectively managing their substance use [10–12]. Overall, the field must develop a stronger understanding of the impact delivering peer services has on worker's professional and personal lives, and how this impact might vary by service setting context.

The PRSS workforce comprises both certified and non-certified peers who work in paid or volunteer positions to deliver a range of support along the continuum from harm reduction to abstinence-focused recovery [13]. It is important to note that people with lived experience have been involved in supporting those who use substances since the beginning of mutual-aid groups (e.g., Alcoholics Anonymous, Narcotics Anonymous, Medication Assisted Recovery Anonymous). However, while peers are involved in sponsorship activities through these mutual support groups, positions of this sort should not be considered PRSS because they exist outside a formal paid or volunteer work environment [14]. People with lived experience have also been highly represented among treatment professionals like addiction counselors [13,15] and, while such experience may be helpful for their work, they do not interact with participants in a peer capacity. The development of PRSS as a profession can be traced to 1999, when Georgia became the first state to allow peer support as a billable provider type for both mental and

behavioral health [15]. As of 2019, 39 US states offered reimbursement for peer services, with training and certification requirements that typically include a specified recovery time, a criminal background check, varied training and exams, and continuing education or recertification [15,16]. Various professional organizations and state-level boards approve these certifications, with as many as 45 distinct categories of certified peers eligible for Medicaid reimbursement [5,16]. This lack of standardization for PRSS certification has generated confusion regarding certified peers' minimal required training and education, role, and scope of work [17].

Understanding workforce outcomes for PRSS is essential for supporting this growing field and ensuring peers' continued wellness and professional growth. These outcomes encompass a wide variety of factors related to peer employment experiences that include *burnout*, *job satisfaction*, *role clarity*, *secondary trauma*, *turnover*, and *absent/presenteeism* [18–20]. The relationship between workplace context and workforce outcomes is well-supported within health professional literature. For example, burnout among health care workers is associated with perceptions of inequity within their organization, perceived job support, supervisory support, and workload [21,22]. Previous reviews have noted high burnout potential among that PRSS workforce due to emotionally laborious conditions stemming from such factors as role ambiguity, limited resources, difficulties establishing boundaries, and vicarious trauma exposure [8,15]. These PRSS outcomes may be moderated by individual characteristics such as coping skills and personal recovery orientation (e.g., abstinence-only vs. harm reduction),) but may also be influenced by workplace factors like belongingness or supervisory support [23–25]. Likewise, it is worthwhile to understand the extent to which peers' well-being both mediates and is mediated by workforce outcomes [26].

The COVID-19 pandemic likely exacerbated factors that can lead to negative peer workforce outcomes. With the sharp increase in drug overdose deaths that started during the pandemic [27], peers report greater stress than ever in their roles [28]. Research notes a high potential for 'dual trauma' during this time, as peers faced pandemic stressors in their personal lives and recovery while simultaneously supporting a population at high risk for adversity and death [25]. These compounding factors make it critical to better understand how peer workplace conditions may contribute to negative outcomes currently associated with this workforce.

Given the rapid expansion of peer support services, publishing a scoping review protocol provides guidance that is of value to this developing area of inquiry. Specifically, outlining the review's rationale can begin the process of establishing new avenues of questioning without having to wait for the often-lengthy review process to result in a final publication. This specific protocol establishes the importance of studying workforce outcomes among peer support workers as it pertains to the quickly evolving field of recovery science [29], and may serve to accelerate the identification of critical workforce issues that are vital for supporting peer workers and improving recovery outcomes across diverse settings. Furthermore, publishing of review protocols aligns with best practices in open science, enabling timely feedback, collaboration, and reduced duplication of efforts [30]. Protocol publication has also been noted to increase the transparency and quality of reporting in the final review [31]. Finally, early protocol dissemination also allows other researchers to adapt or build upon the methodological framework, helping to steer future investigations in meaningful directions [30,32]. Scoping reviews are valuable for analyzing emerging evidence, especially as it remains uncertain whether more focused questions can be formulated regarding the peer workforce [33]. While less intensive than a systematic review, scoping reviews are more rigorous than narrative reviews, which rely on an author's individual expert knowledge [34]. As aligned with scoping review goals to identify the state of knowledge related to an emerging topic area [35], general questions guiding the proposed review will include:

1. What is known about workforce-related outcomes for peers working in the substance use field?

2. What is known about how the structure of work impacts these outcomes?

3. How do these outcomes differ by service setting type?

This proposed effort is unique in its focus from prior published reviews of the PRSS experience or effectiveness by targeting how the *context* of a workplace impacts PRSS outcomes and how these outcomes might vary by workplace type (e.g., clinical, harm reduction settings). Additionally, the proposed review will explore individual-level characteristics of peers (e.g., demographics, training, attitudes) that may moderate workforce outcomes will be explored. We will also explore workforce outcomes as potential mediators of peers' personal recovery outcomes. A preliminary search of MEDLINE, the Cochrane Database of Systematic Reviews, and Joanna Briggs Institute (JBI) Evidence Synthesis was conducted and no current or underway scoping reviews on this topic were identified.

## Methods

We will conduct the proposed scoping review according to frameworks provided by Arksey and O'Malley, Westphaln and colleagues, and Mak and Thomas [35–37]. Results will be reported according to the Preferred Reporting Items for Systematic Reviews and Meta-Analyses extension for scoping reviews (PRISMA-ScR), and we have preregistered the review on Open Science Framework (OSF DOI: https://doi.org/10.17605/OSF.IO/C9YNR). The following describes the methodology of the review according to PRISMA-P (extension for systematic review protocols) standards (see S1 Checklist).

### Eligibility criteria

We will assess peer-reviewed and grey literature describing the experiences of peers participating in substance use disorder PRSS and harm reduction efforts across a variety of workplace settings. PRSS is defined as care delivered by someone who has similar lived experience as the target population [38]. For this review, the term 'peer' is inclusive of individuals in recovery from an SUD who have state or organizational certification, those in recovery without certification, and people who currently use drugs (PWUD). Quantitative and qualitative study designs will be included. We include studies that capture workforce outcomes experienced by peers and report individual or organizational-level variables that influence these outcomes. We consulted previous reviews of healthcare workforce outcomes to develop a list of workforce outcomes for our search strategy [18–20]. Corresponding with the advent of formal peer certification, studies will be restricted to those published from 1 January 1999 to 1 August 2023 and only to settings within the United States. We will exclude studies focusing on similar 'sponsorship' positions in mutual aid organizations, which involve bidirectional support relationships outside a supervised context [39]. We will also exclude studies focusing on peer support outside the substance use recovery and harm reduction fields (e.g., peers focusing on mental or physical health issues). Finally, due to potential inaccuracies in translation that may hinder data extraction, we will exclude papers not published in English. Table 1 displays our proposed inclusion and exclusion criteria.

### Search strategy

An information specialist (TG) will lead a literature search targeting APA PsycINFO® (EBSCO), Embase® (EBSCO), CINAHL® (EBSCO), Web of Science™ (Clarivate), and Google

**Table 1. Screening inclusionary and exclusionary criteria.**

| Inclusion | Exclusion |
|---|---|
| Qualitative or quantitative empirical studies | Not published in English |
| United States-based | Only discusses peers who are in 'sponsorship' positions within substance use mutual aid organizations or people with lived experience working in a professional position (e.g., administrator, addiction counselor, social worker, therapist) |
| Discusses peer recovery support services (PRSS) in the area of substance use harm reduction, treatment, or recovery | Discusses peers who work outside the substance use and harm reduction fields (e.g., mental/physical health, etc.) |
| Discusses certified and uncertified peers who are employed or in volunteer positions as well as people who use drugs (PWUD) who serve as peers | |
| Discusses workforce outcomes | |
| Published between 1/1/1999 to 8/1/2023 | |

List of inclusionary and exclusionary criteria for screening identified literature.

Scholar databases. Various subject headings (i.e., MeSH) will be employed based on the queried database. Keywords will include terms related to peers (e.g., peer, people with lived experience), workforce outcomes (e.g., burnout, compassion fatigue), and organizational environments (e.g., workplace, volunteer). The keywords used to form each search string are included in Table 2 below. A full list of search strings by database is included in S1 Appendix.

We will also include grey literature, that is, any non-peer-reviewed documents captured through the search of databases and through the reference lists of documents fitting our inclusion criteria. We will search for documents on websites of US-based organizations with influence within the field of PRSS, including but not limited to a) Recovery Research Institute, b) Addiction Policy Forum, c) Peer Recovery Center of Excellence, d) SAMHSA, e) Faces and Voices of Recovery, f) National Harm Reduction Coalition, and g) Pure Support. Additional organizations will be included if identified through our publication and database searches. Finally, we will review online materials provided by state-level peer certification organizations, as specified by SAMHSA's *State-by-State Directory of Peer Recovery Coaching Training and Certification Programs* [40].

## Study selection

We will use Rayyan [41] and MAXQDA [42] to manage title/abstract and full-text screening, respectively, eliminating duplicates with Rayyan's duplicate detection function. Two independent reviewers will further evaluate titles and abstracts of peer-reviewed articles to determine inclusion based on our eligibility criteria. Citations meeting the eligibility criteria will undergo a second stage, full-text screening by the reviewers. Agreement between the reviewers will be required for inclusion with a third reviewer resolving any disagreements. Level of consensus between reviewers will be assessed by calculating Cohen's Kappa statistic, with values above 0.6 indicating suitable agreement [43]. If scores fall below 0.6, disagreements will be discussed and resolved, Kappa will be recalculated, and the process repeated until greater than 0.6 is achieved. We will utilize the PRISMA flow diagram to document search outcomes and report the rationale for exclusion of articles.

**Table 2. Keywords informing search strings.**

| Peer Terms |
| --- |
| peer recovery coaches |
| peer provider |
| peer support specialist |
| peer support provider |
| peer recovery support specialist |
| peers |
| peer specialists [[PS]] |
| certified peer specialists |
| peer mentors |
| peer mentorship |
| peer-delivered services |
| peer-delivered support |
| peer certification |
| peer workforce |
| peer recovery workforce |
| peer advocacy |
| people with lived experience |
| people with living experience of drug use |
| people with lived and living experience |
| peer worker |
| peer helper |
| peer administration |
| peer in recovery |
| peer-led support groups |
| peer intervention |
| peer engagement |
| peer-delivered support |
| peer coordinator |
| peer in training |
| peer facilitator |
| peer leadership |
| peer certification |

| Workforce Outcomes |
| --- |
| presenteeism |
| absenteeism |
| burnout |
| workload |
| turnover rate |
| retention |
| recruitment |
| job satisfaction |
| secondary trauma |
| vicarious trauma |
| intent to stay/leave |
| role clarity |
| staff sick leave |
| collaborative practice |
| staff mix |

| Organizational Environments |
| --- |
| workforce |
| health labor supply |
| workplace |
| employee |
| personnel |
| volunteer |
| work environment |
| unlicensed personnel |
| staff |
| human resource |

## Data extraction

Once identified for inclusion, articles will be assigned a unique identifying number, then coded, extracted, and compiled using MAXQDA [a qualitative data analysis software], based on previous recommendations for systematic, scoping reviews [37,44]. One member of the research team will conduct data extraction and another team member will check 10% of the articles for consistency of approach. The following will be extracted from each eligible article: a) bibliographic information (publication type, year); b) study location; c) authors' thesis and research objectives; d) sample size; e) sample information, including peer definition and role type; f) study methodology; g) and context and workplace setting (e.g., rehabilitation center, recovery community organization, etc.). In addition, our primary outcomes will be recorded from each eligible article: h) workforce outcomes (e.g., burnout, job satisfaction, vicarious trauma);) i) individual and organizational-level contributors to workforce outcomes, as well as additional outcomes; and j) author conclusions related to the support of peers within recovery and harm reduction organizations to reduce negative workforce-related outcomes. We will pilot the extraction template with an initial five studies, during which we will adjust extracted information based on the content of the articles. The template will undergo continuous review and be revised, as necessary. If additional extraction categories are introduced, already extracted papers will be revisited for a second iteration.

## Data synthesis and presentation

Results will primarily be presented in narrative form, supplemented by a table highlighting major themes and sub-themes which emerged through the effort. Two reviewers will code the articles in MAXQDA utilizing a deductive coding scheme generated from workforce outcomes along with contributors to these outcomes specified in reviews of the healthcare and general workforce [18–20,45,46]. The reviewers will independently code 10% of documents, aiming for a Cohen's Kappa statistic above 0.6 before dividing and independently coding the remaining documents. The analyzed results will then be presented through thematic analysis, with reference to the objectives of our study. Furthermore, we will interpret relationships between synthesized themes and subthemes, as well as the significance of our findings and any identified gaps in knowledge. We will provide an overview of the descriptive variables of the included studies, such as the research method employed, participant characteristics, and other relevant details. In line with previous recommendations for scoping reviews, we will not undertake an evaluation of individual study quality or conduct a risk-of-bias assessment [36,37]. Substantial amendments to this protocol will be described in the final manuscript.

## Discussion

The proposed scoping review will be the first to systematically explore the characteristics of PRSS and its impact on peer workforce outcomes, extracted from the available literature. Research suggests the PRSS workforce experiences a high frequency of negative outcomes, including burnout, vicarious trauma exposure, and difficulties keeping professional barriers with clients [8–11]. Results will identify PRSS across multiple substance use and harm reduction service settings, characterizing the factors that may increase or decrease the risk of these outcomes, and how these factors vary by setting. The proposed study has been registered in OSF prior to submission. Any amendments to the protocol will be made available through the OSF platform. Publication of this protocol aligns with best practices for Open Science, introducing peer review early in the research process and reducing overlap amongst researchers [31,47].

The proposed review process has noted limitations in that it may fail to capture or fully evaluate certain unpublished materials or forthcoming publications. Additionally, ensuring a comprehensive search poses a challenge due to diverse terminologies used to index the PRSS workforce. This review will serve as a foundation for identifying workforce outcomes and potential mediators of peers' personal recovery and health outcomes.

Developing a well-supported workforce is an essential component of the expansion of peer services recently called for by policymakers and researchers [1,2]. However, the scope of research on workforce conditions for peers is poorly understood. Results of this effort could inform development of more supportive contexts across the spectrum of peer work. The proposed review may identify qualities that promote the success of peer workers or supervisors and locate potential avenues for recruitment. In training, identification of workforce issues can inform strategies to address challenges like burnout and boundary setting. In the workplace, organizational design can better support the retention of peers, including developing opportunities for advancement and career mobility. Findings will aid intervention development by clarifying how such interventions should be adapted to various workplace contexts. The proposed review may also contribute to co-design efforts in service settings by highlighting key areas for collaboration between PRSS and service providers (e.g., supervision, training) [48]. Finally, we will identify gaps in the literature and avenues for future research.

## Supporting information

**S1 Checklist. PRISMA-P 2015 checklist.**
(DOCX)

**S1 Appendix. Full search strategies by database.**
(DOCX)

## Acknowledgments

Mona Stivers provided detailed pre-review editing for this manuscript.

## Author Contributions

**Conceptualization:** Dennis P. Watson.

**Methodology:** Tina Griffin.

**Writing – original draft:** Justin S. Bell, Dennis P. Watson.

**Writing – review & editing:** Justin S. Bell, Sierra Castedo de Martell, Emma Sophia Kay, Mary Hawk, Michelle Hudson, Bradley Ray, Dennis P. Watson.

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
