## [Decision Letter · Decision Letter 0]

15 May 2024

PONE-D-24-06583Workforce outcomes among substance use peer supports and their contextual determinants: A scoping review protocolPLOS ONE

Dear Dr. Bell,

Thank you for submitting your manuscript to PLOS ONE. After careful consideration, we feel that it has merit but does not fully meet PLOS ONE’s publication criteria as it currently stands. Therefore, we invite you to submit a revised version of the manuscript that addresses the points raised during the review process.

We look forward to receiving your revised manuscript.

Kind regards,

Giuseppe Tosto, M.D.

Academic Editor

PLOS ONE

This review is supported by the National Institute on Drug Abuse (R33DA045850).

SC is a volunteer board member for a recovery community organization that delivers peer recovery support services. None of the other authors have competing interests to declare. 

We note that one or more of the authors are employed by a commercial company: name of commercial company. 

Reviewers' comments:

Reviewer's Responses to Questions

**Comments to the Author**

1. Does the manuscript provide a valid rationale for the proposed study, with clearly identified and justified research questions?

Reviewer #1: Partly

Reviewer #2: Yes

2. Is the protocol technically sound and planned in a manner that will lead to a meaningful outcome and allow testing the stated hypotheses?

Reviewer #1: Yes

Reviewer #2: Yes

3. Is the methodology feasible and described in sufficient detail to allow the work to be replicable?

Reviewer #1: Yes

Reviewer #2: Yes

4. Have the authors described where all data underlying the findings will be made available when the study is complete?

Reviewer #1: Yes

Reviewer #2: Yes

5. Is the manuscript presented in an intelligible fashion and written in standard English?

Reviewer #1: Yes

Reviewer #2: Yes

6. Review Comments to the Author

You may also provide optional suggestions and comments to authors that they might find helpful in planning their study.

**Reviewer #1:** This is a well-written manuscript describing the protocol the authors will follow to conduct a scoping review to assess workforce outcomes in peer recovery support staff and how workplace structures and settings impact these outcomes.

The introduction and discussion are very thorough and well-written, describing the importance of the work to be done. The topic of workforce outcomes among this understudied population is of public health importance.

The protocol itself is very well-designed for their aims and clearly written. Strengths include clear definitions of the study population and detailed plans for evaluation, extraction, and interpretation of data sources and data. Good assurances are in place for interrater reliability.

The paper would be strengthened by a discussion of scoping reviews, why this approach is necessary for the research question at hand, and some pros and cons compared with other approaches.

Lastly, the manuscript is missing a discussion of what publishing this plan (or protocol) adds to the scientific literature. The authors should make a clear case for why publishing this now, rather than once the work is done, would be valuable to the field(s).

**Reviewer #2:** good work. Nicely written. I recommend to accept the manuscript. The manuscript describes the methods properly.

7. PLOS authors have the option to publish the peer review history of their article (what does this mean?). If published, this will include your full peer review and any attached files.

Reviewer #1: No

Reviewer #2: **Yes: **Sanjeev Sariya

---

## [Author Response · Author response to Decision Letter 0]

27 Jun 2024

Response to Editor and Reviewers: PONE-D-24-06583

We thank the editor and reviewers for their thoughtful suggestions. We have attempted to address each of the concerns below.

Editor’s Comments

Editor’s Comment Response

We have edited the manuscript to meet PLOS ONE’s style guides according to the templates provided, including updating our file names. 

 We have amended our funding statement to clarify the sources of support for the authors. We have included the suggested language. The final funding statement reads,

“This study was supported by funding from the National Institute on Drug Abuse (R33DA045850, MPIs: Watson and McGuire), including funds from the Justice Community Opioid Innovation Network Collaborative cooperative agreement (UG1DA050065; MPIs: Dennis and Grella). The funder provided support in the form of salaries for authors JB, DW, and MH, but did not have any additional role in the study design, data collection and analysis, decision to publish, or preparation of the manuscript. The specific roles of these authors are articulated in the ‘author contributions’ section. All authors work for universities and health or research organizations that are non-profit entities. There was no additional external funding received for this study.”

3. We note that one or more of the authors are employed by a commercial company: name of commercial company. 

 No authors in this study are employed by a commercial company. All institutions with which authors are affiliated are universities or other not-for profit 501(c)(3) organizations. We included the suggested amended statement referring to the funder of the study in our funding statement. We apologize if we are misunderstanding your request and are happy to correct as advised.

4. Please include captions for your Supporting Information files at the end of your manuscript, and update any in-text citations to match accordingly. We have included captions for our supporting information files at the end of the manuscript. In-text citations have been updated to match accordingly. 

Reviewer 1’s Comments

Reviewer’s Comment Response

1. The paper would be strengthened by a discussion of scoping reviews, why this approach is necessary for the research question at hand, and some pros and cons compared with other approaches. 

We have added details and multiple citations explaining our choice to conduct a scoping review in light of the emerging nature of this research, as well as advantages scoping reviews hold over other approaches (i.e., systematic, narrative reviews): 

“Scoping reviews are valuable for analyzing emerging evidence, especially as it remains uncertain whether more focused questions can be formulated regarding the peer workforce [29]. While less intensive than a systematic review, scoping reviews are more rigorous than narrative reviews, which rely on an author’s individual expert knowledge [30]. As aligned with scoping review goals to identify the state of knowledge related to an emerging topic area [29], general questions guiding this review include:” (p. 7)

2. Lastly, the manuscript is missing a discussion of what publishing this plan (or protocol) adds to the scientific literature. The authors should make a clear case for why publishing this now, rather than once the work is done, would be valuable to the field(s). 

We have added details and citations from established best practices regarding the advantages of publishing a scoping review protocol:

“Publication of this protocol aligns with best practices for Open Science, introducing peer review early in the research process and reducing overlap amongst researchers [42,43]” (p. 19)

---

## [Decision Letter · Decision Letter 1]

3 Sep 2024

PONE-D-24-06583R1Workforce outcomes among substance use peer supports and their contextual determinants: A scoping review protocolPLOS ONE

Dear Dr. Bell,

Thank you for submitting your manuscript to PLOS ONE. After careful consideration, we feel that it has merit but does not fully meet PLOS ONE’s publication criteria as it currently stands. Therefore, we invite you to submit a revised version of the manuscript that addresses the points raised during the review process.

We look forward to receiving your revised manuscript.

Kind regards,

Giuseppe Tosto, M.D.

Academic Editor

PLOS ONE

Journal Requirements:

Additional Editor Comments:

Dear Author,

Please read the comments from Reviewer #1 below.

I encourage you to follow up on the criticisms and update your manuscript accordingly.

Thank you

Reviewers' comments:

Reviewer's Responses to Questions

**Comments to the Author**

1. Does the manuscript provide a valid rationale for the proposed study, with clearly identified and justified research questions?

Reviewer #1: Partly

2. Is the protocol technically sound and planned in a manner that will lead to a meaningful outcome and allow testing the stated hypotheses?

Reviewer #1: Yes

3. Is the methodology feasible and described in sufficient detail to allow the work to be replicable?

Reviewer #1: Yes

4. Have the authors described where all data underlying the findings will be made available when the study is complete?

Reviewer #1: Yes

5. Is the manuscript presented in an intelligible fashion and written in standard English?

Reviewer #1: Yes

6. Review Comments to the Author

You may also provide optional suggestions and comments to authors that they might find helpful in planning their study.

Reviewer #1: The authors have directly responded to the prior questions but with a minimal amount of added text. There is still ambiguity regarding critique 2 (The manuscript is missing a discussion of what publishing this plan (or protocol) adds to the scientific literature. The authors should make a clear case for why publishing this now, rather than once the work is done, would be valuable to the field(s).). This discussion should be in the introduction.

In fact, as written, the Abstract "Objectives," Introduction and beginning of the Discussion lead the reader to believe that this publication IS a scoping review, when it is actually a written plan to conduct a scoping review. Changing the tense used throughout from present to future, i.e., "This scoping review is the first" to "This scoping review will be the first" is a first step.

However, more importantly, in the abstract objectives, the Introduction, and the Discussion, they should state clearly that the objective of this manuscript is to describe a protocol for a FUTURE scoping review.

7. PLOS authors have the option to publish the peer review history of their article (what does this mean?). If published, this will include your full peer review and any attached files.

Reviewer #1: No

---

## [Author Response · Author response to Decision Letter 1]

5 Sep 2024

Response to Editor and Reviewers: PONE-D-24-06583

We thank the reviewer for their thoughtful suggestions. We have attempted to address their concerns below.

Reviewer #1’s Comments

1. The authors have directly responded to the prior questions but with a minimal amount of added text. There is still ambiguity regarding critique 2 (The manuscript is missing a discussion of what publishing this plan (or protocol) adds to the scientific literature. The authors should make a clear case for why publishing this now, rather than once the work is done, would be valuable to the field(s).). This discussion should be in the introduction.

In fact, as written, the Abstract "Objectives," Introduction and beginning of the Discussion lead the reader to believe that this publication IS a scoping review, when it is actually a written plan to conduct a scoping review. Changing the tense used throughout from present to future, i.e., "This scoping review is the first" to "This scoping review will be the first" is a first step.

However, more importantly, in the abstract objectives, the Introduction, and the Discussion, they should state clearly that the objective of this manuscript is to describe a protocol for a FUTURE scoping review.

Response:

We have added language to the introduction that directly speaks to the addition of this protocol to the scientific literature, including research on the peer professional workforce: 

“Given the rapid expansion of peer support services, publishing a scoping review protocol provides guidance that is of value to this developing area of inquiry. Specifically, outlining the review’s rationale can begin the process of establishing new avenues of questioning without having to wait for the often-lengthy review process to result in a final publication. This specific protocol establishes the importance of studying workforce outcomes among peer support workers as it pertains to the quickly evolving field of recovery science [29], and may serve to accelerate the identification of critical workforce issues that are vital for supporting peer workers and improving recovery outcomes across diverse settings. Furthermore, publishing of review protocols aligns with best practices in open science, enabling timely feedback, collaboration, and reduced duplication of efforts [30]. Protocol publication has also been noted to increase the transparency and quality of reporting in the final review [31]. Finally, early protocol dissemination also allows other researchers to adapt or build upon the methodological framework, helping to steer future investigations in meaningful directions [30,32].”

Additionally, we have changed tense throughout and qualified our discussion of the review with ‘proposed’ in order to clarify the distinction that the protocol is describing a future scoping review. We edited portions of the abstract and discussion to clarify the specific objectives of the protocol, i.e.,: 

“Publishing this protocol will help accelerate the identification of critical workforce issues, and bolster the transparency and reporting of the final review.” (p. 3)

---

## [Editor Report · Decision Letter 2]

26 Sep 2024

Workforce outcomes among substance use peer supports and their contextual determinants: A scoping review protocol

PONE-D-24-06583R2

Dear Dr. Bell,

We’re pleased to inform you that your manuscript has been judged scientifically suitable for publication and will be formally accepted for publication once it meets all outstanding technical requirements.

Kind regards,

Giuseppe Tosto, M.D.

Academic Editor

PLOS ONE